# Learning to Multitask

**Yu Zhang**[1], **Ying Wei**[2], **Qiang Yang**[1]
[1]HKUST        [2]Tencent AI Lab
yu.zhang.ust@gmail.com judywei@tencent.com qyang@cse.ust.hk

## Abstract

Multitask learning has shown promising performance in many applications and many multitask models have been proposed. In order to identify an effective multitask model for a given multitask problem, we propose a learning framework called Learning to MultiTask (L2MT). To achieve the goal, L2MT exploits historical multitask experience which is organized as a training set consisting of several tuples, each of which contains a multitask problem with multiple tasks, a multitask model, and the relative test error. Based on such training set, L2MT first uses a proposed layerwise graph neural network to learn task embeddings for all the tasks in a multitask problem and then learns an estimation function to estimate the relative test error based on task embeddings and the representation of the multitask model based on a unified formulation. Given a new multitask problem, the estimation function is used to identify a suitable multitask model. Experiments on benchmark datasets show the effectiveness of the proposed L2MT framework.

## 1   Introduction

Multitask learning [9] aims to leverage useful information contained in multiple tasks to help improve the generalization performance of those tasks. In the past decades, many multitask models have been proposed. According to a recent survey [38], these models can be classified into two main categories: feature-based approach and parameter-based approach. The feature-based approach uses data features as the media to share knowledge among tasks and it usually learns a common feature representation for all the tasks. This approach can be further divided into two categories: shallow approach [9, 2, 43] and deep approach [25]. Different from the feature-based approach, the parameter-based approach links different tasks by placing regularizers or Bayesian priors on model parameters to achieve knowledge transfer among tasks. This approach can be further classified into four categories: low-rank approach [1, 28, 17], task clustering approach [18, 20, 15], task relation learning approach [39, 40, 36, 35, 41, 42, 21, 37], and decomposition approach [10, 19, 44, 16].

Given so many multitask models, one important issue is how to choose a good model among them for a given multitask problem. One solution is to do model selection, that is, using cross validation or its variants. One limitation of this solution is that it is computationally heavy considering that each of the candidate models needs to be trained for multiple times.

In this paper, we propose a framework called Learning to MultiTask (L2MT) to solve this issue in a learning-based approach. The main idea of L2MT is to exploit the historical multitask experience to learn how to choose a suitable multitask model for a new multitask problem. To achieve that, the historical multitask experience is represented as a training set consisting of tuples each of which has three entries: a multitask problem, a multitask model, and the relative test error that equals the ratio of the average test error of the multitask model on the multitask problem over that of the single-task learning model. Based on this training set, we propose an end-to-end approach to learn the mapping from both the multitask problem and the multitask model to the relative test error, where we need to determine the representations of the multitask problem and the multitask model. First, a Layerwise Graph Neural Network (LGNN) is proposed to learn the task embedding as the representation of each

task in a multitask problem and by aggregating of all the task embeddings, the task embedding matrix is used as the representation of the multitask problem. For multitask models which have a unified formulation, task covariance matrices are used as their representations since they play an important role to reveal pairwise task relations. Then both representations of the multitask problem and model are encoded in an estimation function to estimate the relative test error. For a new multitask problem, we can obtain the task embedding matrix via the LGNN learned on the training set and then in order to achieve a low relative test error, we minimize the estimation function to learn the task covariance matrix as well as the corresponding multitask model. Experiments on benchmark datasets show the effectiveness of the proposed L2MT framework.

## 2 A Unified Formulation for Multitask Learning

Before presenting the L2MT framework, in this section, we give a unified formulation for multitask learning by extending that proposed in the survey [38].

Suppose that we are given a multitask problem consisting of $m$ tasks $\{\mathcal{T}_i\}_{i=1}^m$. For task $\mathcal{T}_i$, its training dataset contains $n_i$ data points $\{\mathbf{x}_{i,j}\}_{j=1}^{n_i}$ as well as their labels $\{y_{i,j}\}_{j=1}^{n_i}$, where $\mathbf{x}_{i,j} \in \mathbb{R}^d$ and $y_{i,j} \in \{-1, 1\}$ for classification problems. The learning function for task $\mathcal{T}_i$ is defined as $f_i(\mathbf{x}) = \mathbf{w}_i^T \mathbf{x} + b_i$. A regularized formulation to learn task relations, which can unify several representative models [14, 13, 18, 28, 36, 39, 29, 42, 37], is formulated as

$$\min_{\mathbf{W}, \mathbf{b}, \mathbf{\Omega} \succeq \mathbf{0}} \quad \sum_{i=1}^m \frac{1}{n_i} \sum_{j=1}^{n_i} l\left(\mathbf{w}_i^T \mathbf{x}_{i,j} + b_i, y_{i,j}\right) + \frac{\lambda_1}{2} \text{tr}(\mathbf{W}\mathbf{\Omega}^{-1}\mathbf{W}^T) + \lambda_2 g(\mathbf{\Omega}), \tag{1}$$

where $\mathbf{W} = (\mathbf{w}_1, \ldots, \mathbf{w}_m)$, $\mathbf{b} = (b_1, \ldots, b_m)^T$, $l(\cdot, \cdot)$ denotes a loss function such as the cross-entropy loss, $\mathbf{\Omega} \succeq \mathbf{0}$ means that $\mathbf{\Omega}$ is positive semidefinite (PSD), $\text{tr}(\cdot)$ denotes the trace of a square matrix, $\mathbf{\Omega}^{-1}$ denotes the inverse or pseduoinverse of a square matrix, and $\lambda_1, \lambda_2$ are regularization hyperparameters to control the trade-off among the three terms in problem (1). The first term in problem (1) measures the empirical loss. The second term is a regularizer on $\mathbf{W}$ based on $\mathbf{\Omega}$. According to [39], $\mathbf{\Omega}$, the task covariance matrix, is used to describe pairwise task relations. The function $g(\cdot)$ in problem (1) can be considered as a regularizer on $\mathbf{\Omega}$ to characterize its structure.

The survey [38] has shown that the models proposed in [14, 13, 18, 28, 36, 39, 29, 42, 37] can be formulated as problem (1) with different $g(\cdot)$'s, where the detailed connections between these works and problem (1) are put in the supplementary material for completeness. In the following, we propose two main extensions to enrich problem (1).

Firstly, in Theorem 1, we prove that the Schatten norm regularization is an instance of problem (1). As its special case, the trace norm is widely used in multitask learning [28] as a regularizer to capture the low-rank structure in $\mathbf{W}$. Here we generalize it to the Schatten $a$-norm denoted by $\|\|\cdot\|\|_a$ for $a > 0$, where $\|\|\cdot\|\|_1$ is just the trace norm. To see the relation between the Schatten norm regularization and problem (1), we prove the following theorem with the proof in the supplementary material.

**Theorem 1** *When $g(\mathbf{\Omega}) = \text{tr}(\mathbf{\Omega}^r)$ for any given positive scalar $r$, by defining $\hat{r} = 2r/(r+1)$ and $\lambda_r = (1 + 1/r)(\lambda_1^r \lambda_2 r/2^r)^{1/(r+1)}$, problem (1) reduces to the following problem*

$$\min_{\mathbf{W}, \mathbf{b}} \sum_{i=1}^m \frac{1}{n_i} \sum_{j=1}^{n_i} l\left(\mathbf{w}_i^T \mathbf{x}_{i,j} + b_i, y_{i,j}\right) + \lambda_r \|\|\mathbf{W}\|\|_{\hat{r}}^{\hat{r}}. \tag{2}$$

When $r = 1$, Theorem 1 implies that problem (1) with $g(\mathbf{\Omega}) = \text{tr}(\mathbf{\Omega})$ is equivalent to the trace norm regularization. Even though $r$ can be any positive scalar, problem (2) corresponds to the Schatten $\hat{r}$-norm regularization with $\hat{r} = \frac{2r}{r+1} < 2$ and $\hat{r} \geq 1$ when $r \geq 1$.

Secondly, in the following theorem, we prove that the *squared* Schatten norm regularization is an instance of problem (1).

**Theorem 2** *By defining $g(\mathbf{\Omega}) = \begin{cases} 0 & \text{if } \text{tr}(\mathbf{\Omega}^r) \leq 1 \\ +\infty & \text{otherwise} \end{cases}$, which is an extended real-value function and corresponds to a constraint on $\mathbf{\Omega}$, for any given positive scalar $r$ and $\hat{r} = \frac{2r}{r+1}$, problem (1) is equivalent to the following problem: $\min_{\mathbf{W}, \mathbf{b}} \sum_{i=1}^m \frac{1}{n_i} \sum_{j=1}^{n_i} l\left(\mathbf{w}_i^T \mathbf{x}_{i,j} + b_i, y_{i,j}\right) + \lambda_1 \|\|\mathbf{W}\|\|_{\hat{r}}^2.$*

The aforementioned multitask models with different instantiations of $g(\cdot)$ are summarized in Table 1 in the supplementary material. Based on the above discussion, we can see that problem (1) can embrace many or even infinite multitask models as $r$ in the (squared) Schatten norm regularization can take an infinite number of values. Given a multitask problem and so many candidate models, the top priority is to choose which model to use. One solution is to try all possible models to find the best one but it is computationally heavy. In the following section, we will give our solution: L2MT.

## 3  Learning to Multitask

In this section, we present the proposed L2MT framework and its associated solution.

### 3.1  The Framework

Recall that the aim of the proposed L2MT framework shown in Figure 1 is to determine a suitable multitask model for a test multitask problem by exploiting historical multitask experience. To achieve this, as a representation of historical multitask experience, the training set of the L2MT framework consists of $q$ tuples $\{(\mathcal{S}_i, \mathcal{M}_i, o_i)\}_{i=1}^q$. $\mathbb{S}$ denotes the space of multitask problems and $\mathcal{S}_i \in \mathbb{S}$ denotes a multitask problem. Each multitask problem $\mathcal{S}_i$ consists of $m_i$ learning tasks each of which is associated with a training dataset, a validation dataset, and a test dataset. As we will see later, the $j$th task in $\mathcal{S}_i$ is represented as a task embedding $\mathbf{e}_j^i \in \mathbb{R}^{\hat{d}}$ via applying the proposed LGNN model onto its training dataset and by aggregating of task embeddings of all the tasks, the task embedding matrix $\mathbf{E}_i = (\mathbf{e}_1^i, \ldots, \mathbf{e}_{m_i}^i)$ will be treated as the representation of the multitask problem $\mathcal{S}_i$. $\mathbb{M}$ denotes the space of multitask models and $\mathcal{M}_i \in \mathbb{M}$ denotes a specific multitask model which is trained on the training datasets in $\mathcal{S}_i$. $\mathcal{M}_i$ can be a discrete index for candidate multitask models or a continuous representation based on model parameters. In this sequel, based on the unified formulation presented in the previous section, $\mathcal{M}_i$ is represented by the task covariance matrix $\boldsymbol{\Omega}_i$ and hence $\mathbb{M}$ is continuous. One reason to choose the task covariance matrix as the representation of a multitask model is that the task covariance matrix is core to problem (1) and once it has been determined, the model parameters $\mathbf{W}$ and $\mathbf{b}$ can easily be obtained. One benefit of choosing a continuous $\mathbb{M}$ is that we can learn a new model out of all the candidate models. $o_i \in \mathbb{R}$ denotes the relative test error $\epsilon_{MTL}/\epsilon_{STL}$, where $\epsilon_{MTL}$ denotes the average test error of the multitask model $\mathcal{M}_i$ on the test datasets of multiple tasks in $\mathcal{S}_i$ and $\epsilon_{STL}$ denotes the average test error of a single-task learning (STL) model which is trained on each task independently. Hence, the training process of the L2MT framework is to learn an estimation function $f(\cdot, \cdot)$ to map from $\{(\mathcal{S}_i, \mathcal{M}_i)\}_{i=1}^q$ or concretely $\{(\mathbf{E}_i, \boldsymbol{\Omega}_i)\}_{i=1}^q$ to $\{v(o_i)\}_{i=1}^q$, where $v(\cdot)$, a link function, transforms $o_i$ to make the estimation easier and will be introduced later. Moreover, based on problem (1), we can see $\boldsymbol{\Omega}$ is a function of hyperparameters $\lambda_1$ and $\lambda_2$ and so is the relative test error. Here we make an assumption that $\boldsymbol{\Omega}$ is sufficient to estimate the relative test error. This assumption empirically works very well and it can simplify the design of the estimation function. Moreover, under this assumption, we do not need to find the best hyperparameters for each training tuple, which can save a lot of computational cost.

In the test process, suppose that we are given a test multitask problem $\tilde{\mathcal{S}}$ which is not in the training set. Each task in $\tilde{\mathcal{S}}$ also has a training dataset, a validation dataset and a test dataset. To obtain the relative test error $\tilde{o}$ as low as possible, we resort to minimizing $\gamma_1 f(\tilde{\mathbf{E}}, \boldsymbol{\Omega})$ with respect to $\boldsymbol{\Omega}$ to find the optimal task covariance matrix $\tilde{\boldsymbol{\Omega}}$, where $\tilde{\mathbf{E}}$ denotes the task embedding matrix for the test multitask problem and $\gamma_1$ is a parameter in the link function $v(\cdot)$ to control its monotonic property, and then by incorporating $\tilde{\boldsymbol{\Omega}}$ into problem (1) without manually specifying $g(\cdot)$, we can learn the optimal $\tilde{\mathbf{W}}$ and $\tilde{\mathbf{b}}$ which are used to make prediction on the test datasets.

There are some learning paradigms related to the L2MT framework, including multitask learning, transfer learning [27], lifelong learning [12], and learning to transfer [33]. However, there exist significant differences between the L2MT framework and these related paradigms. In multitask learning, the training set contains only one multitask problem, i.e., $\mathcal{S}_1$, and its goal is to learn model parameters in a given multitask model. The difference between transfer learning and L2MT is similar to that between multitask learning and L2MT. Lifelong learning can be viewed as online transfer/multitask learning and hence it is different from L2MT. Different from learning to transfer which is for transfer learning and relies on handcrafted task features, the proposed L2MT is end-to-end based on neural networks for multitask learning.

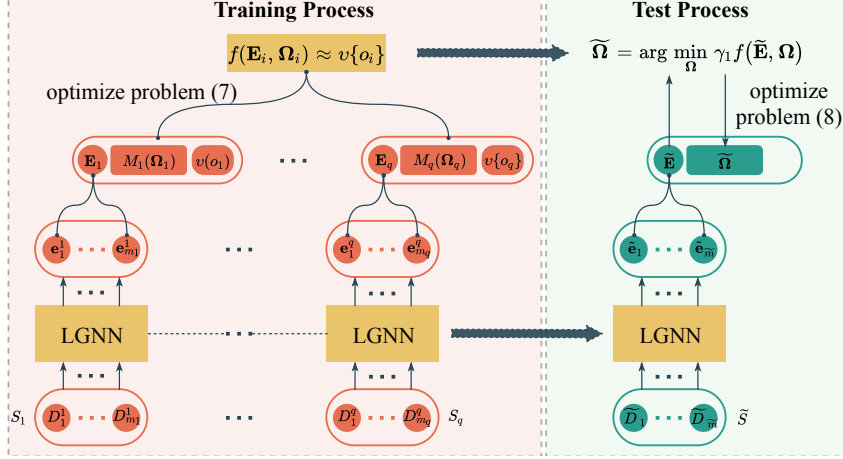

Figure 1: An illustration of the L2MT framework consisting of two stages. The training stage is to learn the estimation function $f(\cdot, \cdot)$ to approximate the relative test error based on multitask problems and multitask models and the test process is to learn the task covariance matrix by minimizing the relative test error or approximately $\gamma_1 f(\tilde{\mathbf{E}}, \mathbf{\Omega})$ with respect to $\mathbf{\Omega}$. $\mathcal{D}_j^i$ denotes the training dataset for the $j$th task in the $i$th multitask problem $\mathcal{S}_i$ and $\tilde{\mathcal{D}}_i$ denotes the training dataset for the $i$th task in the test multitask problem $\tilde{S}$. LGNN, which receives a training dataset as the input and is learned in the training process, is shared by all the tasks in the training and test multitask problems and we plot multiple copies for clear presentation.

## 3.2 Task Embedding

In order to learn the estimation function in the training process, the first thing we need to do is to determine the representation of multitask problems $\{\mathcal{S}_i\}$. Usually each multitask problem is associated with multiple training datasets each of which corresponds to a task. So we can reduce representing a multitask problem to representing the training dataset of a task, which is called the task embedding, in the multitask problem. In the following, we propose a method to represent the task embedding based on neural networks with powerful capacities.

For the ease of presentation, the training dataset of a task in a multitask problem consists of $n$ data-label pairs $\{(\mathbf{x}_j, y_j)\}_{j=1}^n$ by omitting the task index, where $\mathbf{x}_j$ is assumed to have a vectorized representation. Due to varying nature of training datasets in different tasks (e.g., the size and the relations among training data points), it is difficult to use conventional neural networks such as convolutional neural networks or recurrent neural networks to represent a dataset. For a dataset, usually we can represent it as a graph where each vertex corresponds to a data point and the edge between vertices implies the relation between the corresponding data points. Based on the graph representation, we propose the LGNN to obtain the task embedding. Specifically, the input to the LGNN is a data matrix $\mathbf{X} = (\mathbf{x}_1, \ldots, \mathbf{x}_n)$. By using ReLU as the activation function, the output of the first hidden layer in LGNN is

$$\mathbf{H}_1 = \text{ReLU}(\mathbf{L}_1^T \mathbf{X} + \boldsymbol{\beta}_1 \mathbf{1}^T), \tag{3}$$

where $\hat{d}$ denotes the dimension of hidden representations, $\mathbf{L}_1 \in \mathbb{R}^{d \times \hat{d}}$ and $\boldsymbol{\beta}_1 \in \mathbb{R}^{\hat{d}}$ denote the transformation matrix and bias, and $\mathbf{1}$ denotes a vector or matrix of all ones with the size depending on the context. According to Eq. (3), $\mathbf{H}_1$ contains the hidden representations for all the training data points in this task. With an adjacency matrix $\mathbf{G} \in \mathbb{R}^{n \times n}$ to model the relations between each pair of training data points, the output of the $i$th hidden layer ($2 \leq i \leq s$) in the LGNN is defined as

$$\mathbf{H}_i = \text{ReLU}(\mathbf{L}_i^T \mathbf{X} + \mathbf{H}_{i-1} \mathbf{G} + \boldsymbol{\beta}_i \mathbf{1}^T), \tag{4}$$

where $\mathbf{L}_i \in \mathbb{R}^{d \times \hat{d}}$ and $\boldsymbol{\beta}_i \in \mathbb{R}^{\hat{d}}$ are the transformation matrix and bias, and $s$ denotes the total number of hidden layers. According to Eq. (4), the hidden representations of all the data points at the $i$th layer (i.e., $\mathbf{H}_i$) rely on those in the previous layer (i.e., $\mathbf{H}_{i-1}$) and if $\mathbf{x}_i$ and $\mathbf{x}_j$ are correlated according to $\mathbf{G}$ (i.e., $g_{ij} \neq 0$), their hidden representations are correlated. The term $\mathbf{L}_i^T \mathbf{X}$ in Eq. (4) not only preserves the comprehensive information encoded in the original representation but also alleviates

the gradient vanishing issue by achieving the skip connection as in the highway network [32] when $s$ is large. The task embedding of this task, as a result, takes the average of the last hidden layer $\mathbf{H}_s$ over all data points, i.e., $\mathbf{e} = \mathbf{H}_s \mathbf{1}/n$. One advantage of the mean function used here is that it can handle datasets with varying sizes. In LGNN, $\{\mathbf{L}_i\}$ and $\{\boldsymbol{\beta}_i\}$ are learnable parameters based on the objective function presented in the next section.

The graph $\mathbf{G}$ plays an important role in LGNN. Here we use the label information in the training dataset to construct it. For example, when each learning task is a classification problem, $g_{ij}$, the $(i, j)$th entry in $\mathbf{G}$, is set to be 1 when $y_i$ equals $y_j$, $-1$ when $y_i \neq y_j$ and $\mathbf{x}_i$ is one of the $k$ nearest neighbors of $\mathbf{x}_j$ or $\mathbf{x}_j$ is one of the $k$ nearest neighbors of $\mathbf{x}_i$, and 0 otherwise. Based on the definition of $g_{ij}$ and Eq. (4), when two data points are in the same class, their hidden representations have positive effects to each other. When two data points are in different classes and they are nearby (i.e., in the neighborhood), their hidden representations have negative effects to each other.

The original graph neural network [30] needs to solve the fixed point of a recursive equation, which restricts the functional form of the activation function. Graph convolutional neural networks [8, 26, 4] focus on how to select neighbored data points to do the convolution operation, while LGNN aggregates all the neighborhood information in a layerwise manner.

Given a multitask problem consisting of $m$ tasks, we construct a LGNN for all tasks with the shared parameters. Therefore, the task embedding matrix $\mathbf{E} = (\mathbf{e}_1, \ldots, \mathbf{e}_m)$, where $\mathbf{e}_i$ denotes the task embedding for the $i$th task, is treated as the representation for the entire multitask problem. In the next section, we show how to learn the estimation function based on such representation.

### 3.3 Training Process

Recall that the training set in L2MT contains $q$ tuples $\{(\mathcal{S}_i, \mathcal{M}_i, o_i)\}_{i=1}^q$. Applying the LGNN in the previous section, we represent $\mathcal{S}_i$ with $m_i$ tasks as a task embedding matrix $\mathbf{E}_i \in \mathbb{R}^{\hat{d} \times m_i}$. Based on the unified formulation in Section 2, $\mathcal{M}_i$ is represented by the task covariance matrix $\boldsymbol{\Omega}_i \in \mathbb{R}^{m_i \times m_i}$. In the training process, we aim to learn an estimation function mapping from both the task embedding matrix and the task covariance matrix to the relative test error, i.e., $f(\mathbf{E}_i, \boldsymbol{\Omega}_i) \approx \upsilon(o_i)$ for $i = 1, \ldots, q$, where $\upsilon(\cdot)$ is defined as a link function to transform the target. Considering the difficulty of designing a numerically stable $f(\cdot, \cdot)$ to meet all positive $o_i$'s, we introduce the link function, $\upsilon(\cdot)$, which transforms $o_i$ to real scalars being positive or negative. Different $\boldsymbol{\Omega}_i$'s may have variable scales as they are produced by different multitask models with different $g(\cdot)$'s. To make their scales comparable, we impose a restriction that $\mathrm{tr}(\boldsymbol{\Omega}_i)$ equals 1. If some $\boldsymbol{\Omega}_i$ does not satisfy this requirement, we simply preprocess it via $\boldsymbol{\Omega}_i/\mathrm{tr}(\boldsymbol{\Omega}_i)$. Note that different $\mathbf{E}_i$'s can have different sizes as $m_i$ is not fixed. By taking this into consideration, we design an estimation function, where the number of parameters is independent of $m_i$, as

$$f(\mathbf{E}_i, \boldsymbol{\Omega}_i) = \alpha_1 \mathrm{tr}(\mathbf{E}_i^T \mathbf{E}_i \boldsymbol{\Omega}_i) + \alpha_2 \mathrm{tr}(\mathbf{K}_i \boldsymbol{\Omega}_i) + \alpha_4 \mathrm{tr}(\boldsymbol{\Omega}_i^2), \tag{5}$$

where $\mathbf{e}_j^i$ is the $j$th column in $\mathbf{E}_i$, $\mathbf{K}_i$ is an $m_i \times m_i$ matrix with its $(j, k)$th entry equal to $\exp\{-\|\alpha_3(\mathbf{e}_j^i - \mathbf{e}_k^i)\|_2^2\}$, and $\boldsymbol{\alpha} = (\alpha_1, \alpha_2, \alpha_3, \alpha_4)$ contains four real parameters to be optimized in the estimation function. In the right-hand side of Eq. (5), $\mathbf{E}_i^T \mathbf{E}_i$ and $\mathbf{K}_i$ are linear and RBF kernel matrices to define task similarities based on task embeddings, respectively. The first two terms in $f(\cdot, \cdot)$ define the consistency between kernel matrices and $\boldsymbol{\Omega}_i$ with $\alpha_1$ and $\alpha_2$ controlling the positive/negative magnitude to estimate $o_i$. The resultant kernel matrices with the same size as $\boldsymbol{\Omega}_i$ are also the key to empower the estimation function to accommodate $\boldsymbol{\Omega}_i$'s of different sizes.

The link function takes the following form: $\upsilon(o) = \tanh(\gamma_1 o + \gamma_2)$, where $\tanh(\cdot)$ denotes the hyperbolic tangent function to transform a positive $o$ to the range $(-1, 1)$ and $\boldsymbol{\gamma} = (\gamma_1, \gamma_2)$ contains two learnable parameters.

The objective function in the training process is formulated as

$$\min_{\boldsymbol{\Theta}} \frac{1}{q} \sum_{i=1}^q |f(\mathbf{E}_i, \boldsymbol{\Omega}_i) - \upsilon(o_i)| + \lambda \sum_{i=1}^s \|\mathbf{L}_i\|_F^2, \tag{6}$$

where $\boldsymbol{\Theta} = \{\{\mathbf{L}_i\}, \{\boldsymbol{\beta}_i\}, \boldsymbol{\alpha}, \boldsymbol{\gamma}\}$ denotes the set of parameters to be optimized. Here we use the absolute loss as it is robust to outliers. Problem (6) indicates that the proposed method is end-to-end from the training datasets of a multitask problem to its relative test error. We optimize problem (6) via the Adam optimizer in the tensorflow package. In each batch, we randomly choose a tuple (e.g., the $k$th tuple) and optimize problem (6) by replacing the first term with $|f(\mathbf{E}_k, \boldsymbol{\Omega}_k) - \upsilon(o_k)|$ as an approximation. The left part of Figure 1 illustrates the training process.

## 3.4 Test Process

In the test process, suppose that we are given a new test multitask problem $\tilde{\mathcal{S}}$ consisting of $\tilde{m}$ tasks each of which is associated with a training dataset, a validation dataset and a test dataset. The goal here is to learn the optimal $\tilde{\boldsymbol{\Omega}}$ automatically via the estimation function and the training datasets without manually specifying the form of $g(\cdot)$ in problem (1). With $\tilde{\boldsymbol{\Omega}}$ injected, the validation datasets in all tasks can be used to identify the regularization hyperparameter $\lambda_1$ in problem (1) and the test datasets are used to evaluate the performance of L2MT as usual.

For the training datasets in the $\tilde{m}$ tasks, we first apply the learned LGNN in the training process to obtain their task embedding matrix $\tilde{\mathbf{E}} \in \mathbb{R}^{\hat{d} \times \tilde{m}}$. Here the task covariance matrix is unknown and what we need to do is to estimate the task covariance matrix by minimizing the relative test error, which, however, is difficult to measure based on the training datasets. Recall that the estimation function is an approximation of the transformed relative test error by the link function. So we resort to optimize the estimation function instead. Due to the monotonically increasing property of the hyperbolic tangent function used in the link function $\upsilon(\cdot)$, minimizing the relative test error can be approximated by minimizing/maximizing the estimation function when $\gamma_1$ is positive/negative,[1] leading to the minimization of $\gamma_1 f(\tilde{\mathbf{E}}, \boldsymbol{\Omega})$ with respect to $\boldsymbol{\Omega}$, which based on Eq. (5) can be simplified as

$$\min_{\boldsymbol{\Omega}} \rho \operatorname{tr}(\boldsymbol{\Omega}^2) + \operatorname{tr}(\boldsymbol{\Phi}\boldsymbol{\Omega}) \quad \text{s.t. } \boldsymbol{\Omega} \succeq \mathbf{0}, \ \operatorname{tr}(\boldsymbol{\Omega}) = 1, \tag{7}$$

where $\rho = \gamma_1 \alpha_4$, $\tilde{\mathbf{e}}_i$ denotes the $i$th column in $\tilde{\mathbf{E}}$, $\tilde{\mathbf{K}}$ is an $\tilde{m} \times \tilde{m}$ matrix with its $(i,j)$th entry equal to $\exp\{-\|\alpha_3(\tilde{\mathbf{e}}_i - \tilde{\mathbf{e}}_j)\|_2^2\}$, and $\boldsymbol{\Phi} = \gamma_1(\alpha_1 \tilde{\mathbf{E}}^T \tilde{\mathbf{E}} + \alpha_2 \tilde{\mathbf{K}})$. The constraints in problem (7) are due to the requirement that the trace of the PSD task covariance matrix equals 1 as preprocessed in the training process. It is easy to find that problem (7) is convex when $\rho \geq 0$ and otherwise non-convex. Even though the convex/non-convex nature of problem (7) varies with $\rho$, we can always find its efficient solutions summarized in the following theorem.

**Theorem 3** *Define the eigendecomposition of $\boldsymbol{\Phi}$ as $\boldsymbol{\Phi} = \mathbf{U}\boldsymbol{\Lambda}\mathbf{U}^T$ where $\boldsymbol{\Lambda} = \operatorname{diag}(\boldsymbol{\kappa})$ denotes the diagonal eigenvalue matrix with $\boldsymbol{\kappa} = (\kappa_1, \dots, \kappa_{\tilde{m}})^T$ ($\kappa_1 \geq \dots \geq \kappa_{\tilde{m}}$), $\mathbf{U} = (\mathbf{u}_1, \dots, \mathbf{u}_{\tilde{m}})$ denotes the eigenvector matrix, and the multiplicity of $\kappa_{\tilde{m}}$ is assumed to be $t$ ($t \geq 1$). When $\rho = 0$, the optimal solution $\tilde{\boldsymbol{\Omega}}$ of problem (7) is in the convex hull of $\mathbf{u}_{\tilde{m}-t+1}\mathbf{u}_{\tilde{m}-t+1}^T, \dots, \mathbf{u}_{\tilde{m}}\mathbf{u}_{\tilde{m}}^T$. When $\rho < 0$, optimal solutions of problem (7) are in a set $\{\mathbf{u}_{\tilde{m}-t+1}\mathbf{u}_{\tilde{m}-t+1}^T, \dots, \mathbf{u}_{\tilde{m}}\mathbf{u}_{\tilde{m}}^T\}$. When $\rho > 0$, the optimal solution is $\tilde{\boldsymbol{\Omega}} = \mathbf{U}\operatorname{diag}(\boldsymbol{\mu})\mathbf{U}^T$ where $\boldsymbol{\mu}$ is the solution of the following problem*

$$\min_{\boldsymbol{\mu}} \rho\|\boldsymbol{\mu}\|_2^2 + \boldsymbol{\mu}^T\boldsymbol{\kappa} \quad \text{s.t. } \boldsymbol{\mu} \geq \mathbf{0}, \ \boldsymbol{\mu}^T\mathbf{1} = 1. \tag{8}$$

According to Theorem 3, we need to solve problem (8) when $\rho > 0$. Based on the Lagrange multiplier method, we design an efficient algorithm with $O(\tilde{m})$ complexity in the supplementary material. After learning $\tilde{\boldsymbol{\Omega}}$ according to Theorem 3, we can plug $\tilde{\boldsymbol{\Omega}}$ into problem (1) and learn the optimal $\tilde{\mathbf{W}}$ and $\tilde{\mathbf{b}}$ for the $\tilde{m}$ tasks involved in the test multitask problem. The right part of Figure 1 illustrates the test process.

## 3.5 Analysis

The training process of L2MT induces a new learning problem where each multitask problem is used to predict the relative test error and the task embedding matrix contains meta features to describe the multitask problem. In this section, we study the generalization bound for this learning problem.

For the ease of presentation, we assume each multitask problem contains the same number of tasks. By following [7], tasks originate in a common environment $\eta$, which is by definition a probability measure on a learning task. In L2MT, the absolute loss is used and here we generalize it to a general case where the loss function $\bar{l} : \mathbb{R} \times \mathbb{R} \to [0, 1]$ is assumed to be 1-Lipschitz in the first argument. In the test process, we can see that the task covariance matrix is a function of the task embedding matrix and inspired by that we make an assumption that there is some function to represent the task covariance matrix in terms of the task embedding matrix. Based on such assumption, the estimation function is denoted by $\bar{f}(\mathbf{E}) \equiv f(\mathbf{E}, \boldsymbol{\Omega})$. Then the expected loss is defined as $\mathcal{E} = \mathbb{E}[\bar{l}(\bar{f}(\mathbf{E}), \upsilon(o))]$ where the expectation $\mathbb{E}$ is on the space of multitask problems and relative test errors, and $\mathbf{E}$ denotes the task embedding matrix induced by the corresponding multitask problem. The training loss is

defined as $\hat{\mathcal{E}} = \frac{1}{q} \sum_{i=1}^{q} \bar{l}(\bar{f}(\mathbf{E}_i), \upsilon(o_i))$. Based on the Gaussian average [6, 22], we can bound $\mathcal{E}$ in terms of $\hat{\mathcal{E}}$ as follows.

**Theorem 4** *Let $\bar{F}$ be a real-valued function class on the space of task embeddings, the members of $\bar{F}$ have values in $[0, 1]$, and $\mathcal{H}$ denotes the space of transformation functions in LGNN. With probability greater than $1 - \delta$, for any $\bar{f} \in \bar{F}$ and any $h \in \mathcal{H}$, we have*

$$\mathcal{E} \leq \hat{\mathcal{E}} + \frac{c_1 LG(\{\mathbf{E}_i\})}{q} + \frac{c_2 Q \sup_{h \in \mathcal{H}} \|\mathbf{E}\|_F}{q} + \sqrt{\frac{9 \ln(2/\delta)}{2q}},$$

*where $c_1, c_2$ are universal constants, functions in $\bar{F}$ are assumed to have a Lipschitz constant at most $L$, $G(Y) = \mathbb{E} \sup_{\mathbf{y} \in Y} \langle \boldsymbol{\sigma}, \mathbf{y} \rangle$ denotes the Gaussian average where $\boldsymbol{\sigma}$ denotes a generic vector or matrix of independent standard normal variables, $\min_{\mathbf{E}} G(F(\mathbf{E}))$ is assumed to be 0 by following [23], $\| \cdot \|_F$ denote the Frobenius norm, and $Q = \sup_{\substack{\mathbf{E}, \mathbf{E}' \\ \mathbf{E} \neq \mathbf{E}'}} \mathbb{E} \sup_{\bar{f} \in \bar{F}} \frac{\langle \boldsymbol{\sigma}, \bar{f}(\mathbf{E}) - \bar{f}(\mathbf{E}') \rangle}{\|\mathbf{E} - \mathbf{E}'\|_F}$.*

According to Theorem 4, we can see that the expected loss can be upper-bounded by the sum of the training loss, the model complexity based on the task embedding matrices, and a confidence term with the rate of convergence $O(q^{-\frac{1}{2}})$. The Gaussian average on the task embedding matrices induced by LGNN can be estimated via the chain rule [22].

## 4    Experiments

Four datasets are used in the experiments, including the MIT-Indoor-Scene, Caltech256, 20newsgroup, and RCV1 datasets. The MIT-Indoor-Scene and Caltech256 datasets are for image classification, while the 20newsgroup and RCV1 datasets are for text classification. For these two image datasets, we use the FC8 layer of the VGG-19 network [31] pretrained on the ImageNet dataset as the feature extractor. The two text datasets are represented using "bag-of-words", thereby lying in high-dimensional spaces. To reduce the heavy computational cost induced, we preprocess these two datasets to reduce the dimension to 1,000 by following [34] which utilizes ridge regression to select important features. The RCV1 dataset is highly imbalanced as the number of data points per class varies from 5 to 130,426. To reduce the effect of imbalanced classes to multitask learning, we keep the categories whose numbers of data samples are between 400 and 5,000.

Based on each aforementioned dataset , we construct the training set for L2MT in the following two steps: 1) We first construct a multitask problem where each task is a binary classification task. The number of tasks in a multitask problem is uniformly distributed between 4 and 8 as the number of tasks in real applications is limited. For a multitask problem with $m$ tasks, we just randomly sample $m$ pairs of classes along with their data where each task is to distinguish between each pair of classes. 2) We sample $q$ multi-task problems to constitute the training set for L2MT. The test set for L2MT can be obtained similarly and its construction is exclusively different from the training set.

Baseline methods in the comparison consist of a single-task learner (STL), which is trained on each task independently by adopting the cross-entropy loss, and all the instantiation models of problem (1), including regularized multitask learning (RMTL) [14], Schatten norm regularization with $r = 1$ (SNR$_1$) which is the trace norm regularization [28], Schatten norm regularization with $r = 2$ (SNR$_2$) which is equivalent to the Schatten $\frac{4}{3}$-norm regularization according to Theorem 1, the MTRL method [39, 42], squared Schatten norm regularization with $r = 2$ (SSNR$_2$) which is equivalent to squared Schatten $\frac{4}{3}$-norm regularization according to Theorem 2, clustered multitask learning (CMTL) [18], multitask learning with graphical Lasso (glMTL) [36, 29], asymmetric multitask learning (AMTL) [21], and SPATS [37]. So in total there are 10 baseline methods. Moreover, to ensure fairness of comparison, we also allow each baseline method to access and include all training datasets of all training multitask problems, besides the training datasets in a test multitask problem at hand. Consequently, we report the better performance of each baseline method when it learns on the test multitask problem only and on all the training and test multitask problems, respectively.

Usually collecting a training set with many multitask problems needs to take much time and in the experiments, we only collect 100 multitask problems for training where 30% data in each task form the training dataset. For better training on such training set and controlling the model complexity, different $\mathbf{L}_i$'s and $\boldsymbol{\beta}_i$'s ($2 \leq i \leq s$) are constrained to be identical in Eq. (4), i.e., $\mathbf{L}_2 = \ldots = \mathbf{L}_s$ and $\boldsymbol{\beta}_2 = \ldots = \boldsymbol{\beta}_s$. There are 50 test multitask problems in the test set.

Each entry in $\{\mathbf{L}_i\}$ is initialized to be normally distributed with zero mean and variance of $1/100$, and the biases $\{\boldsymbol{\beta}_i\}$ are initialized to be zero. $\boldsymbol{\alpha}$ in the estimation function is initialized to $[1,1,1,0.1]^T$ and $\boldsymbol{\gamma}$ in the link function is initialized to $[1,0]^T$. The learning rate linearly decays from 0.01 with respect to the number of epoches.

To investigate the effect of the size of the training dataset on the performance, we vary the size of training data from 30% to 50% at an interval of 10% with the validation proportion fixed to 30% in the test process and plot the average relative test errors of different methods over STL in Figure 2, where $\tilde{\epsilon}_{MTL} = \frac{1}{\tilde{q}}\sum_{i=1}^{\tilde{q}}\frac{1}{\tilde{m}_i}\sum_{j=1}^{\tilde{m}_i}\tilde{\epsilon}_{i,j}^{MTL}$ denotes the average test error of a multitask model over all the tasks in all the test multitask problems, $\tilde{\epsilon}_{STL}$ has a similar definition for STL, and the average relative test error is defined as $\tilde{\epsilon}_{MTL}/\tilde{\epsilon}_{STL}$. All the relative test errors of STL are equal to 1, and the performance of RMTL is not very good as its assumption that all the tasks

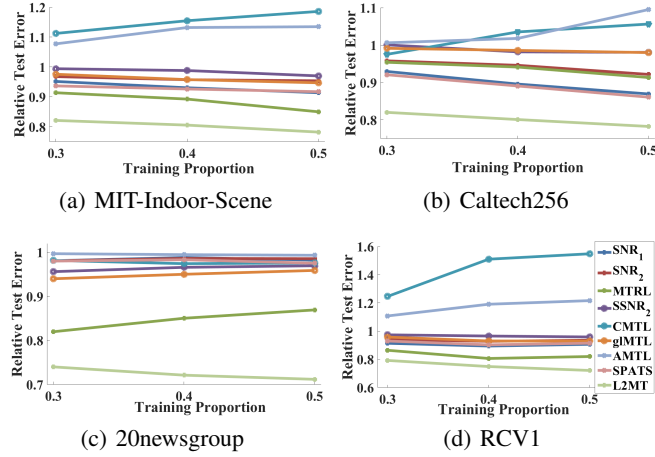

(a) MIT-Indoor-Scene      (b) Caltech256

(c) 20newsgroup      (d) RCV1

Figure 2: Results of different models on four datasets when varying the size of training data.

are equally similar to each other is usually violated in real applications. Hence we omit these two methods in Figure 2 for clear presentation. According to Figure 2, we can see that some multitask models perform worse than STL with relative test errors larger than 1, which can be explained by the mismatch between data and model assumptions imposed on the task covariance. By learning the task covariance directly from data without explicit assumptions, the proposed L2MT performs better than all the baseline methods under different settings, which demonstrates the effectiveness of L2MT.

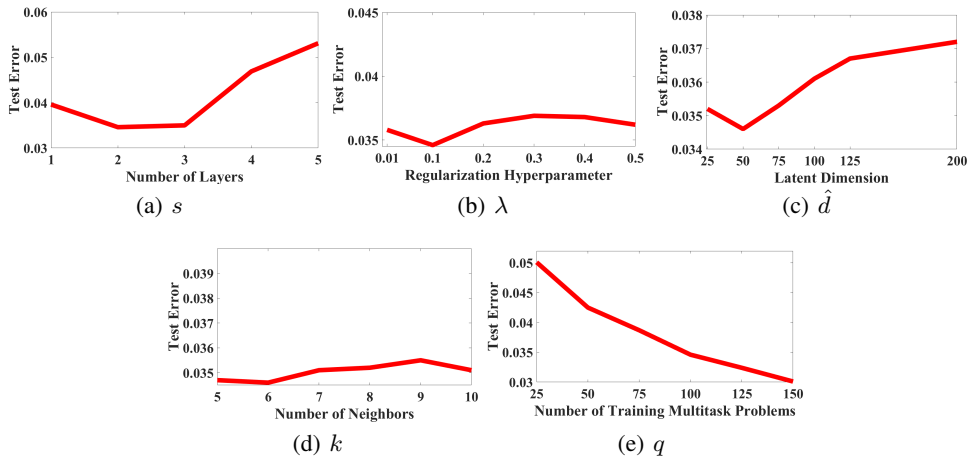

(a) $s$      (b) $\lambda$      (c) $\hat{d}$

(d) $k$      (e) $q$

Figure 3: Sensitivity analysis of L2MT on the 20newsgroup dataset when using 30% data for training.

In Figure 3, we conduct the sensitivity analysis on the 20newsgroup dataset with respect to hyperparameters in L2MT, including the number of layers $s$, the regularization hyperparameter $\lambda$, the latent dimension $\hat{d}$, and the number of neighbors $k$ in LGNN, to see their effects on the performance. According to Figure 3(a), we can see that the performance when $s$ equals 2 and 3 is better than that with $s = 1$, which demonstrates the usefulness of the graph information used in LGNN to learn the task embeddings. Yet the performance degrades when $s$ increases further with one reason that L2MT

is likely to overfit given a limited number of training multitask problems. As implied by Figures 3(b) and 3(d), when $\lambda$ is in $[0.01, 0.5]$ and $k$ in $[5, 10]$, the performance is not so sensitive that the choices are easier and hence in experiments we always set $\lambda$ and $k$ to 0.1 and 6. According to Figure 3(c), when $\hat{d}$ is not very large, the performance is better than that corresponding to a larger $\hat{d}$ where the overfitting is likely to occur. Based on such observation, $\hat{d}$ is set to be 50. Moreover, in Figure 3(e) we test the performance of L2MT by varying $q$, the size of training multitask problems, and we can see that the test error of L2MT decreases when $q$ is increasing, which matches the generalization bound in Theorem 4.

In previous experiments, the VGG-19 network is used as the feature extractor. It is worth noting that L2MT can even be used to update the VGG-19 network. On the Caltech256 and MIT-Indoor-Scene datasets, we use problem (6) as the objective function to fine-tune parameters in the fully connected layers of the VGG-19 network. After fine-tuning, the average test errors of L2MT are reduced by about 5% compared to L2MT without fine-tuning, which demonstrates the effectiveness of L2MT on not only improving the performance of multitask problems but also learning good features.

We also study other formulations for the estimation and link functions. For example, another choice for the estimation function we used is $f(\mathbf{E}, \mathbf{\Omega}) = \alpha_1 \text{tr}(\text{ReLU}(\hat{\mathbf{E}}^T \hat{\mathbf{E}}) \mathbf{\Omega}) + \alpha_2 \text{tr}(\mathbf{\Omega}^2)$ where $\hat{\mathbf{E}} = \mathbf{LE} + \boldsymbol{\beta} \mathbf{1}^T$ with parameters $\mathbf{L}$ and $\boldsymbol{\beta}$, and that for the link function is $\upsilon(o) = \ln(\exp\{\gamma_1\}o + \exp\{\gamma_2\})$, where $\ln(\cdot)$ denotes the logarithm function with base $e$. Compared with the estimation and link functions proposed in Section 3.3, these new functions lead to slightly worse performance (about 2% relative increase on the test error), which demonstrates the effectiveness of the proposed functions over the newly defined ones.

To assess the quality of the learned task covariance matrices by different models, we conduct a case study by constructing a multitask problem consisting of three tasks from the Caltech256 dataset. The first task is to classify between classes 'Bat' and 'Clutter', the second one is to distinguish between classes 'Bear' and 'Clutter', and the last task does classification between classes 'Dog' and 'Clutter'. The learned task correlation matrices, which can be computed from task covariance matrices, by $\text{SNR}_1$, MTRL and L2MT are
$$\begin{pmatrix} 1.0000 & 0.0028 & 0.0789 \\ 0.0028 & 1.0000 & 0.0633 \\ 0.0789 & 0.0633 & 1.0000 \end{pmatrix}, \begin{pmatrix} 1.0000 & -0.0067 & 0.0553 \\ -0.0067 & 1.0000 & 0.0473 \\ 0.0553 & 0.0473 & 1.0000 \end{pmatrix},$$
and
$$\begin{pmatrix} 1.0000 & 0.0057 & 0.0052 \\ 0.0057 & 1.0000 & -0.9782 \\ 0.0052 & -0.9782 & 1.0000 \end{pmatrix}.$$
From the three task correlation matrices, we can see that the correlations between the first and second tasks are close to 0 in the three models, which matches the intuition that bats and bears are almost irrelevant as they belong to different species. The same observation holds for the first and third tasks. The difference among the three methods lies in the correlation between the second and third tasks. Specifically, in $\text{SNR}_1$ and MTRL, those correlations are close to 0, indicating that these two tasks are nearly uncorrelated, and hence the knowledge shared among the three tasks is very limited for $\text{SNR}_1$ and MTRL. On the contrary, in L2MT, the second and third tasks have a highly negative correlation and hence there is strong knowledge leverage between those two tasks, which may be one reason that L2MT outperforms $\text{SNR}_1$ and MTRL.

## 5 Conclusions

In this paper, we propose L2MT to identify a good multitask model for a multitask problem based on historical multitask problems. To achieve this, we propose an end-to-end procedure, which employs the LGNN to learn task embedding matrices for multitask problems and then uses the estimation function to approximate the relative test error. In the test process, given a new multitask problem, minimizing the estimation function leads to the identification of the task covariance matrix. As revealed in the survey [38], there is another representative formulation for the feature-based approach [2, 3, 11] in multitask learning. In our future research, we will extend the proposed L2MT method to learn good feature covariances for multitask problems based on this formulation. Moreover, the proposed L2MT method can be extended to meta learning where LGNN can be used to learn hidden representations for datasets.

## Acknowledgments

This research has been supported by NSFC 61673202, National Grant Fundamental Research (973 Program) of China under Project 2014CB340304, and Hong Kong CERG projects 16211214/16209715/16244616.

## Footnotes

[1] We do not consider a trivial case that $\gamma_1 = 0$ where the estimation function is to approximate a constant.

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
