[Supplementary Material]

# Supplementary Material for "Learning to Multitask"

**Details in the Unified Formulation (1)**

In [14, 13], the priori information about the similarity between a pair of tasks $\mathcal{T}_i$ and $\mathcal{T}_j$ denoted by $s_{ij}$ is used to define a regularizer $\sum_{i=1}^{m} \sum_{j=1}^{m} s_{ij} \|\mathbf{w}_i - \mathbf{w}_j\|_2^2$ to enforce similar tasks to have similar model parameters, where $\|\cdot\|_2$ denotes the $\ell_2$ norm of a vector. It is easy to see that such regularizer equals the second term of problem (1) by setting $g(\mathbf{\Omega}) = \begin{cases} 0 & \text{if } \mathbf{\Omega} = \mathbf{L}_s^{-1} \\ +\infty & \text{otherwise} \end{cases}$, where $\mathbf{L}_s$ is the Laplacian matrix of a graph whose $(i,j)$th entry equals $s_{ij}$. Here $g(\mathbf{\Omega})$, an extended real-value function, acts as a constraint to constrain $\mathbf{\Omega}$ to be $\mathbf{L}_s^{-1}$.

Jacob et al. [18] propose a clustered multitask learning method, which can be viewed as an instance of problem (1), to group all the tasks in the spirit of the $k$-means clustering algorithm by setting $g(\cdot)$ as

$$g(\mathbf{\Omega}) = \begin{cases} 0 & \text{if } \operatorname{tr}(\mathbf{\Omega}) = a, \ b\mathbf{I} \preceq \mathbf{\Omega} \preceq c\mathbf{I} \\ +\infty & \text{otherwise} \end{cases},$$

where $a, b, c$ are additional hyperparameters and $\mathbf{I}$ denotes an identity matrix with appropriate size.

Inspired by the graphical Lasso method [5], we consider an instance of problem (1) by setting $g(\cdot)$ as $g(\mathbf{\Omega}) = \frac{\lambda_1 d}{2\lambda_2} \ln |\mathbf{\Omega}| + \|\mathbf{\Omega}^{-1}\|_1$, where $\|\cdot\|_1$ denotes the $\ell_1$ norm of a vector or matrix, the sum of the absolute values of all entries in it. This setting of $g(\cdot)$ encourages the inverse of $\mathbf{\Omega}$ to be sparse and has been investigated in [36, 29].

Zhang and Yang [37] observe that when there are a large number of tasks, it is better to learn sparse task relations. Then based on problem (1), they aim to learn a sparse $\mathbf{\Omega}$, leading to an implementation of $g(\cdot)$ as $g(\mathbf{\Omega}) = \|\mathbf{\Omega}\|_1$.

In [21], $\mathbf{w}_i$ is assumed to lie in the space spanned by $\mathbf{W}$, i.e., $\mathbf{w}_i \approx \mathbf{W}\mathbf{a}_i$ or equivalently $\mathbf{W} \approx \mathbf{W}\mathbf{A}$, leading to a regularizer $\|\mathbf{W} - \mathbf{W}\mathbf{A}\|_F^2$, where $\|\cdot\|_F$ denote the Frobenius norm. By assuming that the linear spanning $\mathbf{A}$ is sparse, the corresponding $g(\cdot)$ is formulated as

$$g(\mathbf{\Omega}) = \begin{cases} \|\mathbf{A}\|_1 & \text{if } \mathbf{\Omega}^{-1} = (\mathbf{I} - \mathbf{A})(\mathbf{I} - \mathbf{A})^T \\ +\infty & \text{otherwise} \end{cases}. \qquad (9)$$

In Eq. (9), we make several modifications to the original work. Firstly, different tasks are assumed to be equally important. Secondly, to capture the negative correlations between tasks, $\mathbf{A}$ here is allowed to have negative values while in the original work $\mathbf{A}$ is nonnegative. Thirdly, diagonal entries in $\mathbf{A}$ can be zero via the $\ell_1$ regularization to avoid a trivial solution where $\mathbf{A}$ equals $\mathbf{I}$.

The aforementioned multitask models with the corresponding $g(\cdot)$ are summarized in Table 1.

Table 1: Representative multitask models with the corresponding $g(\cdot)$ in problem (1).

| Multitask Model | $g(\cdot)$ |
|---|---|
| [14, 13] | $g(\mathbf{\Omega}) = \begin{cases} 0 & \text{if } \mathbf{\Omega} = \mathbf{L}_s^{-1} \\ +\infty & \text{otherwise} \end{cases}$ |
| [18] | $g(\mathbf{\Omega}) = \begin{cases} 0 & \text{if } \operatorname{tr}(\mathbf{\Omega}) = a, \ b\mathbf{I} \preceq \mathbf{\Omega} \preceq c\mathbf{I} \\ +\infty & \text{otherwise} \end{cases}$ |
| [36, 29] | $g(\mathbf{\Omega}) = \frac{\lambda_1 d}{2\lambda_2} \ln |\mathbf{\Omega}| + \|\mathbf{\Omega}^{-1}\|_1$ |
| [37] | $g(\mathbf{\Omega}) = \|\mathbf{\Omega}\|_1$ |
| Schatten norm regularization | $g(\mathbf{\Omega}) = \operatorname{tr}(\mathbf{\Omega}^r)$ |
| Squared Schatten norm regularization | $g(\mathbf{\Omega}) = \begin{cases} 0 & \text{if } \operatorname{tr}(\mathbf{\Omega}^r) \leq 1 \\ +\infty & \text{otherwise} \end{cases}$ |
| [21] | $g(\mathbf{\Omega}) = \begin{cases} \|\mathbf{A}\|_1 & \text{if } \mathbf{\Omega}^{-1} = (\mathbf{I} - \mathbf{A})(\mathbf{I} - \mathbf{A})^T \\ +\infty & \text{otherwise} \end{cases}$ |

**Proof for Theorem 1**

**Proof.** By setting the derivative of problem (1) with respect to $\mathbf{\Omega}$ to be zero, we can obtain the solution for $\mathbf{\Omega}$ as

$$\mathbf{\Omega} = \left(\frac{\lambda_1}{2\lambda_2 r}\right)^{\frac{1}{r+1}} \left(\mathbf{W}^T\mathbf{W}\right)^{\frac{1}{r+1}}.$$

By plugging this solution into problem (1), we can get an equivalent problem as

$$\min_{\mathbf{W},\mathbf{b}} \sum_{i=1}^{m} \frac{1}{n_i} \sum_{j=1}^{n_i} l\left(\mathbf{w}_i^T\mathbf{x}_j^i + b_i, y_j^i\right) + \lambda_r \mathrm{tr}\left((\mathbf{W}^T\mathbf{W})^{\frac{r}{r+1}}\right).$$

By defining the singular value decomposition (SVD) of $\mathbf{W}$ as $\mathbf{W} = \mathbf{U}_W \mathbf{\Sigma}_W \mathbf{V}_W^T$ where $k$ is the rank of $\mathbf{W}$, $\mathbb{O}^{a\times b}$ denotes the set of orthogonal matrices with size $a \times b$, $\mathbf{U}_W \in \mathbb{O}^{\hat{d}\times k}$, $\mathbf{V}_W \in \mathbb{O}^{m\times k}$, and $\mathbf{\Sigma}_W$ is a $k \times k$ diagonal matrix containing the singular values of $\mathbf{W}$, we have

$$
\begin{aligned}
\mathrm{tr}\left((\mathbf{W}^T\mathbf{W})^{\frac{r}{r+1}}\right) &= \mathrm{tr}\left((\mathbf{V}_W^T\mathbf{\Sigma}_W^2\mathbf{V}_W)^{\frac{r}{r+1}}\right) \\
&= \mathrm{tr}(\mathbf{V}_W^T\mathbf{\Sigma}_W^{\frac{2r}{r+1}}\mathbf{V}_W) \\
&= \mathrm{tr}(\mathbf{\Sigma}_W^{\frac{2r}{r+1}}) \\
&= \|\mathbf{W}\|_{S(\frac{2r}{r+1})}^{\frac{2r}{r+1}},
\end{aligned}
$$

in which we reach the conclusion. $\square$

**Proof for Theorem 2**

**Proof.** The regularizer $R(\mathbf{W})$ is defined as

$$R(\mathbf{W}) = \min_{\mathrm{tr}(\mathbf{\Omega}^r)\leq 1} \mathrm{tr}(\mathbf{\Omega}^{-1}\mathbf{W}^T\mathbf{W}).$$

Since

$$\mathrm{tr}(\mathbf{\Omega}^{-1}\mathbf{W}^T\mathbf{W}) \geq \sum_{i=1}^{m} \frac{\mu_i^2(\mathbf{W})}{\mu_i(\mathbf{\Omega})},$$

where the inequality holds due to the von Neumann's trace inequality, then we can get

$$R(\mathbf{W}) \geq \min_{\mathrm{tr}(\mathbf{\Omega}^r)\leq 1} \sum_{i=1}^{m} \frac{\mu_i^2(\mathbf{W})}{\mu_i(\mathbf{\Omega})} \geq \|\mathbf{W}\|_{S(\hat{r})}^2,$$

where $\mu_i(\cdot)$ denotes the $i$th singular value of a matrix, the second inequality holds due to Lemma 26 in [24], and the equality holds when $\mu_i(\mathbf{\Omega}) = \frac{\mu_i(\mathbf{W})^{\frac{2}{r+1}}}{\left(\sum_j \mu_j(\mathbf{W})^{\frac{2r}{r+1}}\right)^{\frac{1}{r}}}.$ $\square$

**Proof for Theorem 3**

**Proof.** When $\rho > 0$, the Lagrangian of problem (7) is defined as

$$\mathcal{L}(\mathbf{\Omega}, \phi) = \rho\,\mathrm{tr}(\mathbf{\Omega}^2) + \mathrm{tr}(\mathbf{\Phi}\mathbf{\Omega}) - \phi(\mathrm{tr}(\mathbf{\Omega}) - 1),$$

where $\phi$ is the Lagrange multiplier corresponding to the equality constraint. Since $\mathbf{\Omega}$ is PSD, by setting the derivative of $\mathcal{L}(\mathbf{\Omega}, \phi)$ with respect to $\mathbf{\Omega}$ to zero, we can get

$$\tilde{\mathbf{\Omega}} = \max(0, (\phi\mathbf{I} - \mathbf{\Phi})/2\rho),$$

where the $\max$ function operates on the spectral of the matrix. Based on this equation, we can see that $\tilde{\mathbf{\Omega}}$ shares eigenvectors with $\mathbf{\Phi}$ and by plugging this observation into problem (7), it is easy to check that the eigenvalues of $\tilde{\mathbf{\Omega}}$ satisfy problem (8).

When $\rho$ equals 0, based on the Lagrange multiplier method, problem (7) can be reformulated as

$$\min_{\boldsymbol{\Omega}} \max_{\boldsymbol{\Xi} \succeq \mathbf{0}, \phi} \operatorname{tr}(\boldsymbol{\Phi}\boldsymbol{\Omega}) - \operatorname{tr}(\boldsymbol{\Omega}\boldsymbol{\Xi}) - \phi(\operatorname{tr}(\boldsymbol{\Omega}) - 1),$$

which is equal to the dual form as

$$\max_{\boldsymbol{\Xi} \succeq \mathbf{0}, \phi} \min_{\boldsymbol{\Omega}} \operatorname{tr}\left((\boldsymbol{\Phi} - \boldsymbol{\Xi} - \phi\mathbf{I})\boldsymbol{\Omega}\right) + \phi.$$

Since the inner minimization is linear in terms of $\boldsymbol{\Phi}$, the dual form can be simplified as

$$\max_{\boldsymbol{\Xi}, \phi} \phi \quad \text{s.t. } \boldsymbol{\Xi} \succeq \mathbf{0}, \; \boldsymbol{\Xi} = \boldsymbol{\Phi} - \phi\mathbf{I}.$$

It is easy to see that the optimal solution for this dual problem is that $\phi$ equals the minimum eigenvalue of $\boldsymbol{\Phi}$ and $\boldsymbol{\Xi} = \boldsymbol{\Phi} - \phi\mathbf{I}$. So the null space of $\boldsymbol{\Xi}$ is spanned by $\mathbf{u}_{\tilde{m}-t+1}, \ldots, \mathbf{u}_{\tilde{m}}$. Based on the KKT condition, we have $\operatorname{tr}(\tilde{\boldsymbol{\Omega}}\boldsymbol{\Xi}) = 0$ which implies that $\tilde{\boldsymbol{\Omega}}$ is in the null space of $\boldsymbol{\Xi}$, leading to the solution $\tilde{\boldsymbol{\Omega}}$ lying in the convex hull of $\{\mathbf{u}_{\tilde{m}-t+1}\mathbf{u}_{\tilde{m}-t+1}^T, \ldots, \mathbf{u}_{\tilde{m}}\mathbf{u}_{\tilde{m}}^T\}$ which satisfies the equality constraint in problem (7).

When $\rho < 0$, problem (7) is non-convex and we cannot use the Lagrange multiplier method to analyze it. Since the objective function of problem (7) consists of two terms, we can decompose problem (7) into two subproblems:

$$\min_{\boldsymbol{\Omega}} \rho\operatorname{tr}(\boldsymbol{\Omega}^2) \quad \text{s.t. } \boldsymbol{\Omega} \succeq \mathbf{0}, \; \operatorname{tr}(\boldsymbol{\Omega}) = 1, \tag{10}$$

and

$$\min_{\boldsymbol{\Omega}} \operatorname{tr}(\boldsymbol{\Phi}\boldsymbol{\Omega}) \quad \text{s.t. } \boldsymbol{\Omega} \succeq \mathbf{0}, \; \operatorname{tr}(\boldsymbol{\Omega}) = 1. \tag{11}$$

If these two subproblems have some common solution, then this solution will also be the solution to problem (7). Problem (11) is just problem (7) when $\rho$ equals 0 and hence based on the above analysis, its optimal solutions are in the convex hull of $\mathbf{u}_{\tilde{m}-t+1}\mathbf{u}_{\tilde{m}-t+1}^T, \ldots, \mathbf{u}_{\tilde{m}}\mathbf{u}_{\tilde{m}}^T$. As $\rho < 0$, problem (10) is equivalent to the following problem

$$\max_{\boldsymbol{\Omega}} \operatorname{tr}(\boldsymbol{\Omega}^2) \quad \text{s.t. } \boldsymbol{\Omega} \succeq \mathbf{0}, \; \operatorname{tr}(\boldsymbol{\Omega}) = 1,$$

which can be reformulated as

$$\max_{\boldsymbol{\varphi}} \sum_{i=1}^{\tilde{m}} \varphi_i^2 \quad \text{s.t. } \varphi_i \geq 0, \; \sum_{i=1}^{\tilde{m}} \varphi_i = 1, \tag{12}$$

where $\varphi_i$ denotes the $i$th eigenvalue of $\boldsymbol{\Omega}$ and $\boldsymbol{\varphi} = (\varphi_1, \ldots, \varphi_{\tilde{m}})^T$. The equivalence holds since the trace function can be expressed in terms of eigenvalues of a PSD matrix and independent of eigenvectors. For problem (12), we have

$$\sum_{i=1}^{\tilde{m}} \varphi_i^2 \leq \sum_{i=1}^{\tilde{m}} \varphi_i = 1,$$

where the inequality holds since $\varphi_i$ is in $[0, 1]$ implied by the constraints and the equality holds due to the equality constraint in problem (12). So the optimal value for problem (12) is 1, which is achieved when only one entry in $\boldsymbol{\varphi}$ equals 1 while others are 0. It is easy to check that some optimal solutions of problem (11), including $\mathbf{u}_{\tilde{m}-t+1}\mathbf{u}_{\tilde{m}-t+1}^T, \ldots, \mathbf{u}_{\tilde{m}}\mathbf{u}_{\tilde{m}}^T$, satisfy this condition, making them optimal solutions of problem (7). □

**Algorithm for Solving Problem (8)**

Obviously problem (8) is a quadratic program (QP) problem. Many off-the-shelf solvers such as CVX could be used to solve it in polynomial time. To achieve further speedup, we propose a more efficient solution by exploiting the special structure of this problem. Note that the only variable coupling in problem (8) comes from the equality constraint. The Lagrangian corresponding to this constraint is given by

$$\mathcal{L}(\boldsymbol{\mu}, \tau) = \rho\|\boldsymbol{\mu}\|_2^2 + \boldsymbol{\mu}^T\boldsymbol{\kappa} + \tau(\boldsymbol{\mu}^T\mathbf{1} - 1).$$

Setting the derivative of $\mathcal{L}$ with respect to $\mu_i$ to 0, we can see that the minimum is reached when $\mu_i = -\frac{1}{2\rho}(\kappa_i + \tau)$. Since each $\mu_i$ is required to be nonnegative and $\mathcal{L}(\boldsymbol{\mu}, \tau)$ is a quadratic function of $\boldsymbol{\mu}$, the optimal solution for $\mu_i$ is given by

$$\mu_i = \max\left(0, -\frac{\kappa_i + \tau}{2\rho}\right). \tag{13}$$

Plugging the optimal solution of $\mu_i$ into $\mathcal{L}(\boldsymbol{\mu}, \tau)$, we can obtain the dual problem as

$$\min_{\tau} \ 4\rho\tau + \sum_{\tau \le -\kappa_i} (\tau + \kappa_i)^2. \tag{14}$$

Obviously, the objective function of problem (14) is a piecewise linear or quadratic function over regions determined by the sequences $\{-\kappa_i\}$. The main idea of our method is to determine the functional form of problem (14) over each region, then compute the local optimum over each region which has an analytical solution, and finally obtain the global optimum by comparing all the local optima. So the main problem is to determine the coefficients of problem (14) over each region efficiently.

When $\tau \in (-\infty, -\kappa_1]$, the objective function of problem (14) is $c_2\tau^2 + c_1\tau + c_0$, where $c_2 = \tilde{m}$, $c_1 = 2(\sum_{i=1}^{\tilde{m}} \kappa_i + 2\rho)$, and $c_0 = \sum_{i=1}^{\tilde{m}} \kappa_i^2$, and it has an analytical solution as $\tau = \min(-\kappa_1, -\frac{c_1}{2c_2})$. When $\tau \in (-\kappa_{\tilde{m}}, +\infty)$, problem (14) has no well-defined solution since the objective function becomes $4\rho\tau$. So we only need to consider the situation where $\tau \in (-\kappa_1, -\kappa_{\tilde{m}}]$. We summarize the algorithm for solving problem (14) in Algorithm 1. This algorithm needs to scan the sequence $\{\kappa_i\}$ at most twice which costs $O(\tilde{m})$. So the complexity of the whole algorithm is $O(\tilde{m})$ which is much more efficient than existing QP solvers.

---

**Algorithm 1** Algorithm for problem (14)

---

1:  $c_0 := \sum_{i=1}^{\tilde{m}} \kappa_i^2$;  % coefficient for constant term
2:  $c_1 := 2(\sum_{i=1}^{\tilde{m}} \kappa_i + 2\rho)$;  % coefficient for linear term
3:  $c_2 := \tilde{m}$;  % coefficient for quadratic term
4:  $\tau := \min(-\kappa_1, -\frac{c_1}{2c_2})$;
5:  $v := c_0 + c_1\tau + c_2\tau^2$;  % value of current minimum
6:  **for** $i = 2$ to $\tilde{m}$ **do**
7:     % Determine the coefficients over $(-\kappa_{i-1}, -\kappa_i]$;
8:     $c_0 := c_0 - \kappa_i^2$;
9:     $c_1 := c_1 - 2\kappa_i$;
10:    $c_2 := c_2 - 1$;
11:    $\tau_0 := \min(-\kappa_i, \max(-\kappa_{i-1}, -\frac{c_1}{2c_2}))$;
12:    $v_0 := c_0 + c_1\tau_0 + c_2\tau_0^2$;
13:    **if** $v_0 < v$ **then**
14:       $\tau := \tau_0$;
15:       $v := v_0$;
16:    **end if**
17:    $i := i + 1$;
18: **end for**

---

**Proof for Theorem 4**

**Proof**. According to [6], we have

$$\mathcal{E} \le \hat{\mathcal{E}} + \frac{\sqrt{2\pi}}{q} G(S) + \sqrt{\frac{9\ln(2/\delta)}{2q}},$$

where $S = \{\bar{l}(\bar{f}(\mathbf{E}_i), v(o_i)) : \bar{f} \in \bar{F}, h \in \mathcal{H}\}$. By the Lipschitz property of the loss function and Corollary 11 in [23], we have $G(S) \le G(S')$ where $S' = \{\bar{f}(\mathbf{E}_i) : \bar{f} \in \bar{F}, h \in \mathcal{H}\}$. Note that $\mathbf{E}_i$ is defined by $h$. According to Theorem 2 in [22], we have

$$G(S') \le c_1' L G(\{\mathbf{E}_i\}) + c_2' D(\{\mathbf{E}_i\})Q + \min_{\mathbf{E}} G(F(\mathbf{E})),$$

where $c_1', c_2'$ are universal constants, $D(\{\mathbf{E}_i\})$ denotes the diameter among $\{\mathbf{E}_i\}$ and equals the longest distance between any two entries. It is easy to show that $D(\{\mathbf{E}_i\}) \leq 2\sup_{h \in \mathcal{H}} \|\mathbf{E}\|_F$ based on the triangular inequality in the Euclidean distance metric. Since $\min_{\mathbf{E}} G(F(\mathbf{E}))$ is assumed to equal 0, by setting $c_1 = \sqrt{2\pi}c_1'$ and $c_2 = 2\sqrt{2\pi}c_2'$, we reach the conclusion. $\quad\square$

**Statistics for Datasets**

The statistics for the four datasets are recorded in Table 2.

Table 2: Statistics for the four datasets.

| Dataset | # instances | # classes | # instances per class |
|---|---|---|---|
| MIT-Indoor-Scene | 15620 | 67 | [99,734] |
| Caltech256 | 29781 | 256 | [61,800] |
| 20newsgroup | 18774 | 20 | [627,997] |
| RCV1 | 36423 | 21 | [400,5000] |