[Reviews · NeurIPS 2018]

Reviewer 1



This paper proposes a new learning framework to identify effective multitask model for a given multitask problem. This is done by computing task embeddings, using a graph neural network. These embeddings are used to train a function that estimate the relative test error, based on historical multitask experience. Quality: The technical content of the paper is well explained. The experiment section is very dense (probably due to lack of space) and could be better organized with some subsections. Clarity: The paper is generally well-written and structured clearly. The Figures 2 and 3 are way to small and can not be read on a printed version (again, probably due to lack of space). Originality: The main idea of this paper is very interesting and the empirical results look encouraging. I am not convinced by the argumentation given between line 320 and 325. Even if this experiment is interesting, showing that other functions lead to worse performance do not really "demonstrate the effectiveness of the proposed functions"... Significance: The model is evaluated on standard multi-task evaluation benchmarks and compared with many baseline systems. The paper clearly demonstrate the interest of this model for identifying good multitask model for a new multitask problem.

Reviewer 2



This paper presents an end-to-end framework for multitask learning called L2MT. The L2MT is trained on previous multitask problems, and selects the best multitask model for a new multitask problem. Authors also proposed a effective algorithm for selecting the best model on test data. Pros: - This is an clean and straightforward framework for multitask learning, and it is end-to-end. - The authors presented a nice unified formulation for multitask learning, which served as a good motivation for the L2MT framework. Cons: - The 'label' in this learning task is the relative test error of eps_{MTL}/eps_{STL}. In my understanding, eps_{STL} is roughly measuring the level of difficulty of a multitask problem, but is there a better baseline to use here? For instance, can we use the average of STL models as a single model, and use its performance on the test set as a baseline? The reason I am asking is that, eps_{STL} could vary a lot across multitask learning problems, and when eps_{STL} is large, it is not meaningful to predict eps_{MTL}/eps_{STL} accurately, as eps_{MTL} will be large as well. It will be better to have a more robust baseline to replace eps_{STL}. - It is fine to assume that the tasks under a single multitask problem have the same data dimension. But the authors also assume the same data dimension across multitask problems, as there is a single LGNN used across problems. I think this does not always hold if you want to train on historic multitask problems of various kinds, e.g. images of different sizes. - In the experiment, the authors constructed the set of multitask problems by basically bootstrapping from a single dataset, while I would be more interested if the set of multitask problems are from different datasets, and that will make the experiment results more interesting and realistic. - The generalization analysis uses multiple Lipschitz assumptions, some of which are not that obvious to me. For instance, could you please explain why in Theorem 4 you can assume \bar F to be Lipschitz constant? The function \bar f seems really complicated as it involves an implicit relation between the task covariance matrix \Omega and the task embedding matrix E.

Reviewer 3



This paper proposes a learning to multitask (L2MT) framework to identify a suitable multitask model under a unified formulation (1) for a multitask problem. This paper is well written and easy to follow. Different from traditional model selection approaches (e.g., cross validation) to determine the suitable multitask model for a multitask problem, the proposed L2MT provides a novel learning approach to solve this issue. This paper is novel in the following aspects: 1) The proposed LGNN is used to learn a dataset embedding for a task. This is totally different from traditional approaches which rely on manually designed features to represent a dataset. With the LGNN, the training process of L2MT becomes end-to-end, which alleviates the tedious feature engineering process. 2) Based on the unified multitask formulation (1), each multitask model is parameterized by the task covariance matrix $\Omega$. With such continuous parameterization, the testing process can learn a novel $\Omega$ out of all the known multitask models presented in Section 2 and this makes learning a better multitask model than all the existing models possible. I think this is a reason that the proposed L2MT can outperform all baseline models in the experiments. 3) The construction of the estimation function is based on the kernel function, which is easy to understand and also facilitates the optimization in the test process. The interesting idea of introducing a link function to transform labels simplifies the construction of the estimation function. 4) The experiments are thorough by presenting different aspects of the proposed L2MT. I have some minor questions: (1) In Theorems 1 and 2, Schatten a-norm and squared Schatten a-norm with 1<=a<2 are instances of formulation (1). What about other cases for a, e.g., 0=2? Can these cases be unified into formulation (1)? (2) The generalization bound seems important to L2MT. It is better to put the analysis in the main body instead of the supplementary material though I understand this is due to page limit.

Reviewer 4



The authors of the manuscript consider the problem of selecting an information transfer algorithm in the multi-task setting. Instead of performing some kind of model selection, they propose a learning-based approach: based on historical MTL experience they build a mapping from MT training data and MT model into relative test performance improvement that using that MT model brings compared to solving each task in isolation. For a new, test, MTL problem they optimise the resulted function with respect to MT model to maximise the improvement. Pros: This is a natural continuation of multi-task/learning to learn ideas with adding one more layer in the hierarchy that has been mentioned in works of Baxter [7]. The authors' implementation of it relies on a nice unification of various MTL methods presented in section 2. Cons: The practicality of the proposed L2MT seems rather limited to me - MT experiences used for training need to be very similar to the problems appearing at the testing stage. Also, data-embedding LGNN seem to require all experiences to be MTL problems with the same number of dimension. Consequently, in the experimental evaluation both training and test MTL problems come from the same dataset. This, in my opinion, greatly limits applicability of L2MT in realistic scenarios, especially since the number of training experiences has to be significant. It would be interesting to see what happens if test and training problems are coming from different datasets, for example, by merging MIT-Indoor-Scene and Caltech256 experiments. Experiments: - how parameters were selected for the baselines? - how MT models and relative test errors were obtained for the training stage of L2MT? Additional comments: - I believe, in Theorem 4 one needs the MTL experiences to be sampled from a distribution, not individual tasks. Also, without a more direct bound on the Gaussian width of {E_i} the resulting bound is not particularly conclusive - the authors may want to add [1] to the related work, as it is based on a similar idea but for domain adaptation [1] Transfer Learning via Learning to Transfer, Ying WEI, Yu Zhang, Junzhou Huang, Qiang Yang ICML'18 Update: based on the authors' response and other reviews I raise my score to 6